# A study on the determination of the factors affecting the happiness levels of older individuals during the COVID-19 pandemic in Turkish society

Nurşen Çomaklı Duvar[1☺¤a], Ahmet Kamil Kabakuş [2☺¤b], Neslihan İyit[3☺¤c], Ömer Alkan [1,4¤a]*

1 Department of Econometrics, Ataturk University, Erzurum, Türkiye, 2 Department of Management Information Systems, Ataturk University, Erzurum, Türkiye, 3 Department of Statistics, Selçuk University, Konya, Türkiye, 4 Master Araştırma Eğitim ve Danışmanlık Hizmetleri Ltd. Şti., Ata Teknokent, Erzurum, Türkiye

☺ These authors contributed equally to this work.
¤a Current address: Department of Econometrics, Faculty of Economics and Administrative Sciences, Ataturk University, Erzurum, Türkiye
¤b Current address: Department of Management Information Systems, Faculty of Economics and Administrative Sciences, Ataturk University, Erzurum, Türkiye
¤c Current address: Department of Statistics, Faculty of Science, Selçuk University, Konya, Türkiye
* oalkan@atauni.edu.tr

**Citation:** Çomaklı Duvar N, Kabakuş AK, İyit N, Alkan Ö (2025) A study on the determination of the factors affecting the happiness levels of older individuals during the COVID-19 pandemic in Turkish society. PLoS ONE 20(1): e0316000. https://doi.org/10.1371/journal.pone.0316000

**Data Availability Statement:** The data underlying this study is subject to third-party restrictions by the Turkish Statistical Institute. Data area available from the Turkish Statistical Institute (bilgi@tuik.

## Abstract

This study aims to determine the factors affecting the happiness levels of older individuals in Türkiye during the COVID-19 pandemic. The microdata set from the 2020 Life Satisfaction Survey conducted by the Turkish Statistical Institute was utilized, involving 1,863 individuals aged 60 and above. The relationship between happiness levels and various factors was investigated using the chi-square independence test, and the factors affecting happiness were further analyzed through generalized ordered logistic regression. According to the generalized ordered logistic regression model, participants in the 60–64 age group are 10.1% less likely to report happiness compared to those aged 65 and older. Men are 4.3% less likely than women to report happiness. Furthermore, individuals with no formal education and those with primary school education have a 14.4% and 9.4% higher likelihood of happiness, respectively, compared to university graduates. The literature on happiness demonstrates the relationship between different factors and happiness. This study determined that such factors as gender, age, educational status, source of happiness, health satisfaction, hope scale, and homeownership have an impact on the happiness levels of older individuals. The amount of societal support provided to older individuals can be an indicator of their level of happiness.

## Introduction

Happiness is viewed as a positive inner experience that results from cognitive and emotional interpretations of an individual's life [1]. As emphasized in this definition, happiness is a

gov.tr) for researchers who meet the criteria for access to confidential data. The authors of the study did not receive any special privileges in accessing the data.

**Funding:** The author(s) received no specific funding for this work.

**Competing interests:** The authors have declared that no competing interests exist.

comprehensive concept rooted in two components: emotional, which denotes the balance between pleasant and unpleasant emotions, and cognitive, which is related to mental health [2]. It is defined as a positive inner experience resulting from an emotional interpretation of one's life and cognitive function or how much an individual enjoys their life [3].

Happiness is an important factor in healthy ageing [4]. In this context, happiness is an emotional state characterized by feelings of pleasure and satisfaction. Happiness is often associated with morale, satisfaction, well-being, successful ageing, quality of life, and well-being [5]. Considering the results connected to happiness, several studies have suggested that happiness can be used used to treatmental illnesses by increasing hope, improving psychological resistance, and strengthening the defence against stress [6].

Similar to happiness, many studies have shown that people view old age as one of the inevitable events of human life [7]. Elderliness accompanies a decline in physical and mental strength, making older individuals more vulnerable. The "automatic" living characterizing modern life, living in apartment flats or tiny houses, reduced number of children, and children migrating expose the older population to loneliness, depression, anxiety, and ultimately, physical and mental disorders [8]. A region or country is regarded as having an older population if its older adults (those 65 years and older) make up more than 7% of the overall population; when this ratio reaches 20%, it is referred to as being "super-aged" [9]. In developed countries, those aged 65 and older are considered old, while in developing countries, those aged 60 and older are considered old [10]. The number of older individuals globally has increased significantly due to declining birth rates and increasing life expectancy [11]. According to the World Health Organization (WHO) data from 2018, it is estimated that the older adult population will make up 22% of the global population by 2050, with individuals aged 60 and over considered old [12]. According to Turkish Statistical Institute data, it is estimated that in Türkiye, the population aged 65 and over will comprise 20.8% of the population by 2050 [13].

Since the first infected case was detected in Wuhan, China, in December 2019, the COVID-19 pandemic has spread globally, causing an unprecedented public health crisis [14]. Due to its high contagiousness and rapid transmission rate, the World Health Organization declared the COVID-19 epidemic a pandemic on March 11 [15]. The COVID-19 pandemic has created a unique situation for Türkiye and other countries [15, 16]. As in many other countries, the Turkish government implemented nationwide isolation measures to prevent the spread of the COVID-19 virus [17]. Places where social interaction is common, such as cafes, restaurants, gyms, and private education facilities, are closed [18].

According to studies, the COVID-19 outbreak has had the most severe effects on the older population [19–22]. Although the COVID-19 outbreak has affected all age groups, the majority of confirmed cases and deaths have been recorded among older individuals [23]. As stated in a report by the USA Centers for Disease Control and Prevention (CDC) in March 2020, over 80% of deaths occurred in patients aged 65 and older, illustrating their vulnerability to the virus [24, 25]. Older individuals are vulnerable to severe infections and reduced immune function due to ageing and preexisting health conditions [26, 27]. Therefore, more protective measures should be taken for older individuals [28]. Health care, emergency response, and quarantine for older individuals have become essential, especially for those with underlying medical conditions since they are more vulnerable [14]. As in many countries, a curfew was imposed on older individuals [29].

Pandemics also have negative psychosocial effects, such as health anxiety, panic, stress, insomnia, and depression [30]. During the COVID-19 pandemic, factors such as social isolation, stress, and changing lifestyles might have affected the mental state of older individuals [31]. Therefore, the mental health of older individuals necessitates a higher level of attention and care since they constitute the demographic group experiencing the most extended social isolation [14]. Studies conducted prior to the COVID-19 pandemic indicate that older

individuals have mental health issues such as loneliness and hopelessness; therefore, these negative outcomes might have been exacerbated by enforced social isolation [32, 33].

Although there are many studies on the psychological states of the general uninfected population isolated at home during the COVID-19 outbreak, there are very few studies on the vulnerable demographic of older individuals [34, 35]. Little is known about the happiness levels of older people in Türkiye during COVID-19. To the best of our knowledge, this is the first study to determine the factors affecting the happiness levels of older individuals in Türkiye during the pandemic. Unlike other studies in the literature, this study aims to investigate the various factors affecting the happiness levels of older individuals whose lives were altered by strict isolation measures during COVID-19. To achieve this goal, a large dataset is used to estimate the factors affecting the happiness levels of older individuals in Türkiye.

The spread of fear and hatred towards vulnerable older individuals due to viruses has become a significant global social issue. Discriminatory behaviors, such as abandonment or mockery, have emerged [36]. These conditions necessitate identifying and designing appropriate interventions, strategies, and policy measures to create and sustain positive, healthy, and effective mental health conditions [14]. The present study aims to determine the factors affecting the happiness levels of older individuals in Türkiye during the COVID-19 pandemic.

The older adults are at-risk groups during the COVID-19 pandemic. Identifying the factors affecting the happiness levels of older individuals during the COVID-19 period is of great importance for developing more targeted strategies to minimize the adverse effects of the pandemic on the elderly and protect their quality of life. It is important to improve the quality of life of older individuals and provide a better ageing process. COVID-19 has been a disease that carries serious health risks for older individuals. The uncertainties of COVID-19 and health concerns have led to an increase in mental health problems such as anxiety, depression and stress in older adults. To cope with these problems, it is critical to identify the factors that affect happiness and adapt mental health services accordingly. Therefore, identifying the factors that affect the happiness levels of older individuals to protect their physical and psychological health may help reduce the disease's adverse effects.

The findings indicate that several factors affect the happiness levels of those aged 60 and older. According to the research, factors such as gender, age, education level, income level, and being hopeful affect the happiness of older individuals. It is essential to know these factors to design more effective social policy and health services according to the needs of the elderly population. Thus, better health services, social assistance and education programs can be developed for older adults. To better understand the relationship between happiness-related factors, a larger study sample is needed among older individuals whose lives were changed by strict isolation methods during the COVID-19 period in Türkiye. For this reason, the purpose of this study is to review the current policies throughout the country, to re-evaluate the needs that arise in the face of extraordinary altering circumstances such as the COVID-19 pandemic, and to fill this gap in the literature.

Understanding the factors affecting the happiness levels of older individuals during the COVID-19 period can increase the effectiveness of the measures and support mechanisms to be taken to help them overcome this difficult period with less damage and live a healthier and happier old age.

In this study, the research questions focused on the happiness level of older individuals in Türkiye are as follows:

Research Question 1: What were the happiness levels of older individuals in Türkiye according to their sociodemographic and economic characteristics during the COVID-19 pandemic?

Research Question 2: Did the sociodemographic and economic characteristics of older individuals in Türkiye affect their happiness levels during the COVID-19 pandemic?

## Literature review

In recent years, happiness has been the focus of many studies [37]. A Taiwanese research study concludes that happiness is an overall evaluation of life experience and well-being, which is significant to older individuals [38]. Sometimes, happiness is associated with pleasure. According to a study conducted in Thailand, education level and gender are also associated with happiness, psychological well-being, and quality of life [39]. Studies in South Africa suggest that happiness, welfare, and quality of life are influenced by various factors such as health, social life, age, and gender [40–42]. Studies in China also indicated that variables like age, health, income, and religious beliefs are determinants of happiness [43, 44]. Research conducted in the Americas and South Africa found that happiness, well-being, and quality of life are affected by various variables, including health status, marital status, and religious affiliation [41, 45]. According to a study conducted in Malaysia, happiness is substantially associated with higher household income, employment, higher education level, and being a woman [3]. Another study revealed that happiness in Malaysia decreases with age and is notably poor among people aged 50 and over [46].

According to studies conducted on older individuals in Iran, happiness is related to gender, education, socioeconomic position, and health, all of which directly affect happiness [47, 48]. A study on the happiness of older individuals in Israel found that education and income positively affect happiness, while gender and age do not [49]. A survey of the sources of happiness for older individuals in Türkiye determined that age, gender, education level, and income level affect happiness [50]. Research conducted in Brazil indicates that age and psychological factors affect levels of happiness [51]. A study conducted in China determined that gender, income, and education have a significant effect on happiness [52].

The psychosocial and mental health issues caused by COVID-19 in older individuals must be thoroughly and fully discussed [14]. According to a Korean study, adults over the age of 60 require special consideration since they are more physically and intellectually fragile than other age groups [53]. Studies conducted in China and the Philippines indicate that adults who live alone, lack social support, are concerned about the spread of the disease, are in poor health, and are exposed to prejudice were psychologically affected by the pandemic, resulting in increased levels of stress, anxiety, and depression [54–56]. Research in the United States has revealed that older individuals were more likely to be affected by the pandemic for a variety of reasons, including their increased likelihood of having age-related chronic conditions and live alone [23, 57]. According to a European-based study, social deprivation and loneliness have an impact on happiness [58].

Loneliness and social isolation are associated with a decreased ability to perform everyday activities, increased levels of general sickness, and increased morbidity [59]. According to a study conducted in Spain, social isolation, stress, and lifestyle changes experienced by older individuals during the COVID-19 pandemic damaged their mental health [31]. In a study on older individuals coping with the pandemic in the United States, it was found that social isolation increased anxiety accompanied by depressive symptoms, and that depression symptoms was more prevalent among older individuals who felt alone and lacked social support [60]. According to a UK study, social isolation protected older individuals against COVID-19, but it interfered with activities many people enjoy, including socializing with family and friends, volunteering, and engaging in daily physical activity [61]. A study conducted in the United States found that while protection and social distancing prevented older individuals from contracting the virus, the resulting social isolation may have negatively affected their physical and mental health [62].

According to a study conducted in India, happiness is directly related to being hopeful [63]. A study on the factors leading to happiness in Türkiye concluded that happiness is positively

related to income and the degree of hopefulness [64]. Research conducted in China indicates that those who live with family members are happier [65]. In a study conducted in Türkiye during the COVID-19 period, it was determined that the pandemic positively affected hopefulness and that being with family members throughout the pandemic improved happiness [17]. Mental health problems increase susceptibility to infection [66]. Studies conducted before the COVID-19 outbreak in Türkiye indicate that older individuals suffer from mental health issues such as loneliness and hopelessness; therefore, these negative consequences might have been compounded by obligatory social isolation regulations [32, 33]. However, studies among older individuals in Türkiye show that hopelessness levels range from above average to high compared to the general population [33, 67]. Additionally, research suggests that social support and spirituality are essential variables in fostering hope [68, 69]. According to a study conducted in Türkiye, the level of hope is crucial for reducing anxiety and tension in older individuals [17].

## Methods

The methodology details the data sources used, their coding, and weighting, followed by the modelling approach employed to clarify the relationships between variables.

### Study design

In this study, we used the microdata set obtained from the Life Satisfaction Survey (LSS) conducted by the Turkish Statistical Institute in 2020. The primary purpose of the LSS is to measure individual's general perception of happiness, social values, general satisfaction in essential life domains, and satisfaction with public services, as well as to track changes in these levels of satisfaction over time. The LSS was first conducted in 2003 and is conducted annually.

The research scope includes households in all settlements within the borders of Türkiye, excluding those in institutional settings (such as schools, dormitories, hotels, kindergartens, nursing homes, hospitals, and prisons) as well as residents in military barracks and houses. Additionally, settlements with a population comprising less than 1% of the total population (such as small villages, camps, and hamlets), where it is considered impractical to reach the required number of sample households, are also excluded [70].

The research was designed to provide a total estimate for Türkiye. The sampling method utilized in this study is two-stage stratified cluster sampling. In the first stage, clusters (blocks) containing an average of 100 households were selected for the sample. In the second stage, sample addresses were derived from the clusters using the systematic selection method [70].

A total of 541 clusters, including 70 clusters from rural-urban settlements, 65 clusters from rural-rural settlements, and 406 clusters from urban-urban settlements, were selected for the sample in the design created to produce estimates based on Türkiye. The sample comprises 5,410 households, consisting of 700 from rural-urban settlements, 650 from rural-rural settlements, and 4,060 from urban-urban settlements (10 from each cluster). Of these households included in the sample, 4,784 households were interviewed and 10,103 individuals were interviewed from these households in the sample.

### Data

In the LSS, both home questionnaires and individual questionnaires were administered. These data were received in two separate Excel files and merged into a single data file. The weighting process was conducted using selection probabilities derived from the dataset obtained through the multi-stage sample design. The final weights were determined based on combination of many factors. Initial weights were calculated as the inverse of the selection probabilities.

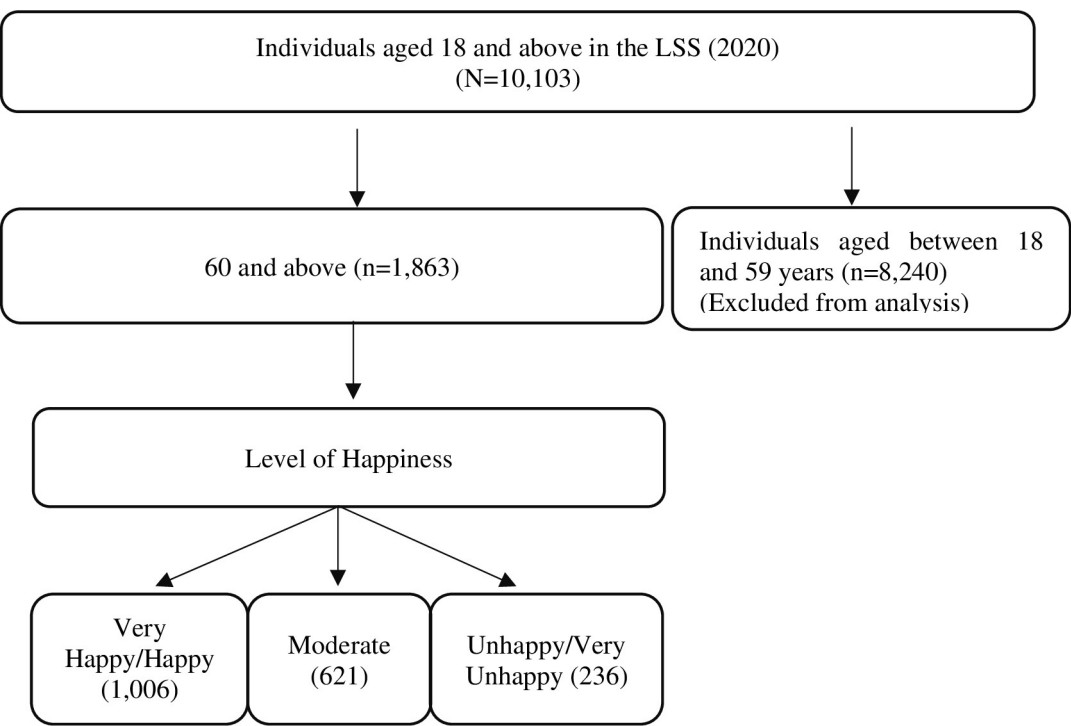

**Fig 1. Sample selection process and determination of dependent variables to assess the factors influencing the happiness levels of older individuals.**

Iterations were performed using the integrated calibration method, which incorporates projected population ratios, to estimate the total population (excluding institutional populations) and the number of households across Türkiye [70].

In this study, we used the data of 1,863 older individuals aged 60 and over who participated in the 2020 LSS. The selection process of the sample included in the study is illustrated in Fig 1.

## Measures and variables

The study's dependent variable is the happiness levels of older individuals, which were measured using the question, "How happy are you when you think about your life as a whole?" The dependent variable is ordinal with three categories: very happy/happy, moderate, and unhappy/very unhappy. Similar measures of happiness levels are used in the literature [49, 50, 52].

The study's independent variables were derived from the variables in the LSS microdata set and are prominent in the relevant literature. In the study, the independent variables thought to affect the happiness levels of older individuals in the COVID-19 pandemic period include the following:

- Age group (60–69, 70–79, 80–89) [46, 53, 71, 72]

- Gender (male, female) [3, 39, 47]

- Education level (did not complete a school, primary school, elementary school, high school, 2 or 3 year college, 4 year college or faculty/master's/doctoral degree) [1, 3, 47, 49]

- Source of happiness (himself/herself/spouse, children/mother/father, friends/nephews/ grand children/other than those mentioned here, all family members)

- Source of happiness (success, Job/money/other than those mentioned here, health, love) [1, 50]

- Health satisfaction (very satisfied/satisfied, moderate, not satisfied/not satisfied at all)

- Hope scale (not hopeful at all, neither hopeful nor not hopeful, hopeful, very hopeful) [17, 32, 33, 39, 63, 64, 68]

- Household size (1–3 people, 4–5 people, 6 and above) [17, 23, 28, 65, 73]

- Homeownership (homeowner, not a homeowner) [50, 64].

## Statistical analysis

Survey statistics in Stata 15 (Stata Corporation) were used to utilized to accommodate the complex sampling design and weights. Weighted analysis was conducted, and frequencies and percentages were obtained according to the happiness levels of older individuals who were the research subjects. The chi-square independence test was employed to explore the relationship between the happiness levels of older individuals and the independent variables. Subsequently, generalized ordered logistic regression analysis was used to identify the factors affecting happiness among older individuals during the COVID-19 pandemic [74]. Bivariate relationships were assessed by examining significant differences using Pearson's chi-square tests for the categorical variables [75].

Regression methods examine the relationship between dependent and explanatory variables. Simple and multiple linear regression methods, which are used when the dependent variable is quantitatively measured, are the most common among them. However, when the dependent variable is categorical, such as in multinomial logistic regression, the assumptions of Ordinary Least Squares method are not met [76]. Alternatively, logistic regression is suitable for predicting a categorical variable from a set of predictor variables. Similar to linear regression, logistic regression is more appropriate when the dependent variable is categorical with two or more unordered (nominal) categories [77].

In logistic regression, the methodology used depends on the number of categories of the qualitative dependent variable and whether these categories are unordered (nominal) or ordered (ordinal). Logistic regression analysis is categorized into binary, multinomial, and ordinal logistic regression based on the nature of the dependent variable [76]. The ordered logit model, also known as the proportional odds model proposed by McCullagh [78], is considered a significant extension of the binary logit model when the dependent variable is measured on an ordinal scale with categorical values [79].

The ordered logistic regression model is written as:

$$Pr(Y \le y_j | x) = \left[ \frac{\exp(a_j - x'\beta)}{1 + \exp(a_j - x'\beta)} \right] j = 1, 2, \ldots, J-1 \qquad 1$$

This equation represents the natural logarithm of the odds ratios of the model, and when written as Eqs 2 and 3:

$$logit\left(\pi_j\right) = \log\left(\frac{\pi_j}{1 - \pi_j}\right) \qquad 2$$

$$log = \left[\frac{Pr(Y \le y_j | x)}{Pr(Y > y_j | x)}\right] = a_j - x'\beta \qquad 3$$

Here, $y_j$ represents the ordered categorical dependent variable, $x'$ denotes the vector of independent (explanatory) variables, $a_j$s are intercepts corresponding to $J$-1 estimators, $a_1 \leq a_2 \leq \cdots \leq a_{j-1}$, βs represent a vector of regression coefficients corresponding to $\beta = (\beta_1, \ldots, \beta_k)'x'$. In ordered logistic regression, the assumption of parallel lines is known as the parallel slopes assumption, asserting that the parameter estimates do not vary across intercepts, testing the adequacy of identical parameters for all categories [80].

When dealing with dependent variables that have an ordinal structure and multiple categories, the ordered logistic regression model is typically employed. This model is suitable when the dependent variable is categorical and ordinal. Alternatively, the generalized ordered logistic model was developed as an extension of the ordered logistic model to address situations where the assumption of parallel curves is not met [81].

## Ethics approval

The data were obtained through the joint teamwork of both the Turkish Statistical Institute (TSI) and the European Union Statistical Office (SOEU). We obtained this data from TSI in return for a contract without needing an ethics committee document and used it in our study.

TSI is an institution that compiles, evaluates, and presents statistical information to decision-makers to prepare development plans and programs, make economic decisions, and address all other issues needed. TSI carries out internationally comparable statistical production activities according to the standards of organizations such as the European Union Statistical Office, the United Nations, OECD, ILO, etc. TSI collects data within the scope of the Official Statistics Program. The Official Statistics Program is prepared for five-year periods based on the Turkish Statistics Law No. 5429 to determine the basic principles and standards regarding the production and publication of official statistics and to ensure the production of up-to-date, reliable, timely, transparent and impartial data in areas of need at national and international levels [82]. TSI also conducts the Türkiye Health Survey within the scope of the Official Statistics Program put into effect by law. Since the Türkiye Health Survey is conducted within the scope of legal responsibility by the state, ethical approval is not required [83].

For this study, secondary data were employed. Official approval was received from the Turkish Statistical Institute to use the microdata set from the Life Satisfaction Survey. The Turkish Statistical Institute also received a "Letter of Undertaking" authorizing it to use the study's data.

The letter of undertaking for the use of micro data without restrictions in dissemination:

Article 1- This letter of undertaking determines the rules, principles and obligations of the use of micro data, which are safe to disclose apart from the Presidency.

Article 2-This letter of undertaking regulates the use of micro data sets of Life Satisfaction Survey, 2020, within the framework of the Directive on Access and Use of Micro Data in line with the purpose specified in Article 1.

Article 3- The following provisions apply for the use of micro data:

a. Findings obtained by the researcher as a result of incorrect calculation only bind the researcher.

b. The researcher refers to the micro data of the Institution that he uses while disclosing the results obtained from the study.

c. The researcher is obliged to send a copy of the published report, article, publication etc. to the Institution Library within three months at the latest. Subsequent micro data usage requests of the researcher who is found not to fulfill this obligation are not covered.

d. The researcher cannot reproduce, give to third parties, sell or transfer the micro data set he obtained.

Article 4-The researcher, by taking into account the principles of confidentiality defined in 13. and 14. articles of Turkish Statistical Institution numbered 5429 and Regulation on Procedures and Principles Regarding Data Confidentiality and Confidential Data Security in Official Statistics, is deemed to guarantee hereby that he shall not disclose the information, table, etc. violating this principle and shall only use micro data for statistical purposes.

## Results

### Descriptive statistics and chi-square tests

Table 1 displays the findings regarding the factors affecting the happiness levels of older individuals in Türkiye during the COVID-19 pandemic.

According to the chi-square independence test results, a significant relationship was found between the happiness levels of older individuals and various socio-demographic and economic factors during the COVID-19 pandemic. Among individuals aged 60–69, 50.4% are very happy/happy, 36.1% are moderately happy, and 13.6% are very unhappy/unhappy. In terms of gender, 54.4% of older female individuals are very happy/happy, while 11.9% are very unhappy/unhappy. For education, 58.5% of older individuals without a degree were very happy/happy, 31.6% were moderately happy, and 12.1% were very unhappy/unhappy. When the source of happiness is the whole family, 58.5% of older individuals are very happy/happy, 32.2% are moderately happy, and 21.9% are very unhappy/unhappy. Finally, 39.3% of older individuals who do not own a house are very happy/happy, and 19.9% are very unhappy/unhappy.

### Model estimation

Using the generalized ordered logistic regression model, the factors affecting the happiness levels of older individuals during the COVID-19 epidemic were determined. According to the approximate likelihood-ratio test of proportionality of odds, the null hypothesis states that "there is no difference in the coefficients between the models." The test revealed that the data do not permit using the proportional odds model, as the test is significant ($\chi2 = 64.68$, $p < 0.001$). Therefore, we did not use a proportional odds model; instead, we employed a generalized ordered logistic regression model. The results obtained from this model are presented in Table 2.

The generalized ordered logistic regression model was found to be statistically significant ($F = 9.05$, $p < 0.001$), explaining 28.7% of sentence severity variance according to the Cox and Snell $R^2$ Index, or 17.6% according to the McFadden Index, which is commonly used in ordered logistic regressions (Following adjustment, the McFadden Adjusted $R^2$ was 14.8%).

To test for multicollinearity among the independent variables included in the study, the "variance inflation factor" (VIF) values were examined [84]. As seen in Table 2, the VIF values are less than 5. This indicates that no variable is responsible for the multicollinearity problems among the variables.

The independent variables in the model were identified by the literature. These variables, found in the LSS, were either nominal (categorical without a specific order) or ordinal (categorical with a specific order). All the variables in the study were qualitative. Ordinal and nominal variables were defined as dummy variables to observe the effects of the categories for all variables included in the generalized ordered logistic regression [85, 86].

As a result of the estimated model, it was checked whether the variables were statistically significant at 1%, 5%, and 10% significance levels. According to the generalized ordered logistic regression model results, there was a significant relationship between the happiness level of

**Table 1. Findings on the factors affecting the happiness levels of the older individuals during the COVID-19 pandemic and chi-square independence test statistics.**

| Variables | | n(%) | Level of Happiness | | | P |
|---|---|---|---|---|---|---|
| | | | Very Happy/ Happy | Moderate | Unhappy/Very Unhappy | |
| **Age** | 60–69 | 1,104 | 556(50.4) | 398(36.1) | 150(13.6) | 0.000[a] |
| | 70–79 | 602 | 353(58.6) | 178(29.6) | 71(11.8) | |
| | 80–89 | 157 | 97(61.8) | 45(28.7) | 15(9.6) | |
| **Gender** | Female | 955 | 539(56.4) | 302(31.6) | 114(11.9) | 0.046[b] |
| | Male | 908 | 467(51.4) | 319(35.1) | 122(13.4) | |
| **Education Level** | Did not complete a school | 535 | 313(58.5) | 157(29.3) | 65(12.1) | 0.002[a] |
| | Primary school | 880 | 486(55.2) | 294(33.4) | 100(11.4) | |
| | Elementary school | 116 | 59(50.9) | 39(33.6) | 18(15.5) | |
| | High school | 151 | 62(41.1) | 58(38.4) | 31(20.5) | |
| | 2 or 3 year college | 74 | 31(41.9) | 33(44.6) | 10(13.5) | |
| | 4 year college or faculty/master's/doctoral degree | 107 | 55(51.4) | 40(37.4) | 12(11.2) | |
| **Source of Happiness as a Person** | Himself/herself/spouse | 160 | 72(45.0) | 53(33.1) | 35(21.9) | 0.000[a] |
| | Children/mother/father | 285 | 129(45.3) | 97(34.0) | 59(20.7) | |
| | Friends/nephews/grand children/other than those mentioned here | 147 | 61(41.5) | 62(42.2) | 24(16.3) | |
| | All family members | 1,271 | 74(58.5) | 409(32.2) | 118(9.3) | |
| **Source of Happiness** | Success | 47 | 26(55.3) | 17(36.2) | 4(8.5) | 0.077[c] |
| | Job/money/other than those mentioned here | 111 | 45(40.5) | 44(39.6) | 22(19.8) | |
| | Health | 1,542 | 846(54.9) | 509(33.0) | 187(12.1) | |
| | Love | 163 | 89(54.6) | 51(31.3) | 23(14.1) | |
| **Health Satisfaction** | Very satisfied/satisfied | 950 | 642(67.6) | 237(24.9) | 71(7.5) | 0.000[a] |
| | Moderate | 540 | 230(42.6) | 251(46.5) | 59(10.9) | |
| | Not satisfied/not satisfied at all | 373 | 134(35.9) | 133(35.7) | 106(28.4) | |
| **Hope Scale** | Not hopeful at all | 119 | 37(31.1) | 32(26.9) | 50(42.0) | 0.000[a] |
| | Not hopeful | 616 | 235(38.1) | 265(43.0) | 116(18.8) | |
| | Neither hopeful nor not hopeful | | 327(58.2) | 194(34.5) | 41(7.3) | |
| | Hopeful | 476 | 331(69.5) | 118(24.8) | 27(5.7) | |
| | Very hopeful | 90 | 76(84.4) | 12(13.3) | 2(2.2) | |
| **Household Size** | 1–3 people | 1,550 | 834(53.8) | 517(33.4) | 199(12.8) | 0.068[c] |
| | 4–5 people | 188 | 98(52.1) | 60(31.9) | 30(16.0) | |
| | 6 people and above | 125 | 74(59.2) | 44(35.2) | 7(5.6) | |
| **Homeownership** | Homeowner | 1,507 | 866(57.5) | 476(31.6) | 165(10.9) | 0.000[a] |
| | Not a homeowner | 356 | 140(39.3) | 145(40.7) | 71(19.9) | |

[a]p < .01
[b]p < .05
[c]p < .10

older individuals and the variables of age, gender, education level, source of happiness, health satisfaction, hope level, number of household members, and homeownership status.

Marginal effects indicate changes in the probability of older individuals' predicted happiness levels for a unit change in the independent variable. Considering the generalized ordered logistic regression model presented in Table 2, the likelihood of survey participants aged 60–69 years being "very happy/happy" is 1.4% lower than that of individuals in the age group of 80–89 years. The probability of men being "very happy/happy" is 3.5% lower than that of women. Individuals who have completed high school have an 8.9% higher likelihood of being

**Table 2. Generalized ordered logistic regression model results and odds ratios.**

| Variables | Generalized Ordered Logistic Regression | | | | | | | | VIF |
|---|---|---|---|---|---|---|---|---|---|
| | Very Happy/Happy | | | | Moderate | | | | |
| | OR | Std. E. | dy/dx | Std. E. | OR | Std. E. | dy/dx | Std. E. | |
| **Age (reference category: 80–89)** | | | | | | | | | |
| 70–79 | 1.658[a] | 0.349 | -0.101[a] | 0.041 | 2.120 | 0.655 | 0.042 | 0.402 | 3.54 |
| 80–89 | 1.188[b] | 0.257 | -0.341 | 0.042 | 1.758 | 0.584 | -0.009 | 0.413 | 3.37 |
| **Gender (reference category: female)** | | | | | | | | | |
| Male | 1.192 | 0.148 | -0.035[b] | 0.021 | 1.195 | 0.211 | 0.019 | 0.024 | 1.18 |
| **Education Level (reference category: did not complete a school)** | | | | | | | | | |
| Primary school | 1.331 | 0.196 | 0.043 | 0.028 | 1.181 | 0.250 | 0.013 | 0.017 | 1.67 |
| Elementary school | 1.419 | 0.381 | 0.026 | 0.052 | 1.610 | 0.562 | 0.043 | 0.034 | 1.25 |
| High school | 2.483[a] | 0.556 | 0.089[b] | 0.044 | 2.504[a] | 0.729 | 0.095[a] | 0.032 | 1.32 |
| 2 or 3 year college | 3.045[a] | 0.903 | 0.153[b] | 0.062 | 2.055[b] | 0.889 | 0.070 | 0.048 | 1.16 |
| 4 year college or faculty/master's/doctoral degree | 1.605[c] | 0.402 | 0.081 | 0.052 | 1.188 | 0.465 | 0.014 | 0.033 | 1.27 |
| **Source of Happiness as a Person (reference category: all family members)** | | | | | | | | | |
| Himself/herself/spouse | 1.337 | 0.272 | -0.059 | 0.041 | 2.202[a] | 0.547 | -0.020 | 0.039 | 1.06 |
| Children/mother/father | 1.401[b] | 0.216 | -0.068[b] | 0.031 | 1.872[a] | 0.395 | 0.008 | 0.031 | 1.11 |
| Friends/nephews/grand children/other | 1.478[c] | 0.318 | -0.079[c] | 0.044 | 1.452 | 0.378 | 0.046 | 0.042 | 1.06 |
| **Source of Happiness (reference category: love)** | | | | | | | | | |
| Success | 1.131 | 0.388 | -0.025 | 0.069 | 0.846 | 0.514 | 0.039 | 0.076 | 1.26 |
| Job/money/other | 2.239[a] | 0.705 | -0.163[a] | 0.062 | 1.905 | 0.746 | 0.093 | 0.063 | 1.60 |
| Health | 1.093 | 0.217 | -0.018 | 0.040 | 0.948 | 0.253 | 0.022 | 0.039 | 1.85 |
| **Health Satisfaction (reference category: not satisfied/not satisfied at all)** | | | | | | | | | |
| Very satisfied/satisfied | 0.292[a] | 0.045 | 0.261[a] | 0.031 | 0.235[a] | 0.045 | -0.105[a] | 0.033 | 1.92 |
| Moderate | 0.828 | 0.134 | 0.040 | 0.034 | 0.354[a] | 0.072 | 0.084[b] | 0.034 | 1.82 |
| **Hope Scale (reference category: not hopeful)** | | | | | | | | | |
| Not hopeful at all | 1.319 | 0.305 | -0.058 | 0.047 | 2.837[a] | 0.634 | -0.110[b] | 0.045 | 1.14 |
| Neither hopeful nor not hopeful | 0.454[a] | 0.062 | 0.172[a] | 0.029 | 0.350[a] | 0.077 | -0.075[b] | 0.029 | 1.37 |
| Hopeful | 0.303[a] | 0.045 | 0.255[a] | 0.030 | 0.322[a] | 0.078 | -0.152[a] | 0.030 | 1.42 |
| Very hopeful | 0.186[a] | 0.064 | 0.343[a] | 0.058 | 0.130[a] | 0.094 | -0.253[a] | 0.057 | 1.13 |
| **Household Size (reference category: (6 people and above)** | | | | | | | | | |
| 1–3 people | 1.350 | 0.316 | -0.060 | 0.046 | 2.246[b] | 1.045 | -0.006 | 0.045 | 2.39 |
| 4–5 people | 1.148 | 0.326 | -0.027 | 0.056 | 2.665[b] | 1.242 | -0.046 | 0.055 | 2.32 |
| **Homeownership (reference category: not a homeowner)** | | | | | | | | | |
| Homeowner | 0.561[a] | 0.079 | 0.118[a] | 0.028 | 0.605[a] | 0.107 | -0.068[b] | 0.028 | 1.04 |
| **Constant** | 1.389 | 0.522 | | | 0.102 | 0.058 | | | |

[a] p < .01

[b] p < .05

[c] p < .10; Std. E.: Standard Error; VIF: Variance Inflation Factor

"very happy/happy" compared to the reference group (those who did not graduate from any school). Those with a two or three-year college degree have a 15.3% higher likelihood of being "very happy/happy" compared to the reference group of high school graduates. The probability of individuals who have completed high school being "moderately happy" is 9.5% higher than that of the reference category.

The likelihood of an individual whose source of happiness is children/parents being "very happy/happy" is 6.8% lower than that of the reference group (entire family). The likelihood of

an individual whose source of happiness is friends/nephews/grandchildren/others being "very happy/happy" is 7.9% lower than that of the reference group. The likelihood of an individual whose primary source of happiness is work/money/others being "very happy/happy" is 16.3% lower than that of the reference group (love).

Those who are very satisfied/satisfied with their health have a 26.1% higher likelihood of being "very happy/happy" compared to the reference category (not satisfied/not satisfied at all). Individuals who are very satisfied/satisfied with their health have a 10.5% lower likelihood of being "moderately happy" compared to those not satisfied/not satisfied at all. Individuals with moderate health satisfaction have an 8.4% higher likelihood of being "moderately happy" compared to the reference category. The probability of an individual who is hopeful being "very happy/happy" is 25.5% and 34.3% higher than the reference group (not hopeful). The probability of an individual not hopeful at all being "moderately happy" is 25.3% lower than the reference category.

A homeowner is 11.8% more likely to be "very happy/happy" than non-homeowners, the reference group. Homeowners are 6.8% less likely to be "moderately happy" than non-homeowners.

## Discussion

The study found that certain variables are associated with happiness. These variables include age, gender, educational status, source of happiness, hope scale, and economic status. Considering conditions that directly affect people, such as the pandemic, inability to socialize, separation from family, and health concerns, participants can be predicted to have a lower level of happiness. Based on the literature review, there is limited research examining the happiness levels of older individuals during the COVID-19 pandemic. By determining the factors affecting the happiness of older individuals in Türkiye, it may be advantageous to pay more attention to older individuals and to direct future research on this topic. This study determined the socio-demographic and economic factors that affect the happiness level of older individuals in Türkiye.

It was determined in this study that age significantly influences the level of happiness. Individuals between 60 and 69 years are more likely to report being "very happy/happy" than older participants. Previous studies consistently show that age significantly impacts life satisfaction [87–90]. Studies have indicated that older participants generally report lower happiness levels [1, 91, 92]. A previous study revealed that while older participants did not differ significantly in happiness levels on specific scales, those aged between 60 and 69 reported lower happiness levels across all measures [67]. Testing the hypothesis of declining happiness with age, the authors of a study involving 7,399 British individuals found no age-related changes in happiness levels; however, they identified that social capital and engagement predict lifelong happiness [93].

Additionally, studies highlight that age adversely affects happiness [65]. Aging deteriorates physical health among older individuals and reduces their overall quality of life. Physical and mental health mutually influence each other, with declining physical health contributing to unhappiness [94].

Another variable associated with the happiness levels of older individuals is gender. Men are less likely to report being "very happy/happy" than women. Similar studies have also indicated that women tend to be happier than men [39, 95, 96]. Consistently across these studies, women have a higher probability of reporting happiness compared to men [97, 98]. However, contrary to this study, some research has suggested that older women may report lower levels of happiness due to higher rates of depression and anxiety compared to men [99, 100].

This study revealed that educational levels are associated with the happiness levels of older individuals. Those who have completed high school or college are more likely to report being "very happy/happy" than those with lower education levels. Additionally, individuals with a high school education are more likely to report being "moderately happy" than those who did not graduate. Similar studies have shown that education influences happiness [39, 101]. Educational attainment is linked to happiness among older adults [102, 103], as higher education levels correlate with better physical and mental health strategies [23, 103, 104]. Previous research consistently reports a positive relationship between education and happiness, with higher education levels associated with greater happiness [91, 105, 106]. Therefore, educational qualifications may relate particularly to economic resources in older age, which in turn are associated with happiness [107].

Another significant variable associated with the happiness levels of older individuals is the source of happiness. This study found that individuals whose primary source of happiness is children/parents are less likely to report being "very happy/happy" compared to those whose source is the entire family. Older individuals whose happiness stems from their entire family are more likely to be happy. As previous studies suggest, family as a source of happiness influences happiness probability [39, 65]. Living with family members, including spouses and children, provides both material and spiritual care and support, fostering love, interest, self-respect, and a sense of worth, leading to a happy and healthy life [1, 108]. Similar studies highlight that the quantity and quality of family support significantly impact older individuals' happiness levels [1, 109, 110]. Other research indicates that while family members are a significant source of instrumental support, friends and others are less likely to provide such support [47, 111]. Family relationships constitute a substantial part of social support for older adults, providing an environment of love and protection. Therefore, family support and relationships can play a vital role in maintaining the psychological and spiritual health of older individuals during the pandemic [14]. Studies show that older individuals in families with low solidarity and functioning experience more depression and loneliness [14, 53].

Another variable related to the happiness levels of older individuals is the primary source influencing their happiness. This study revealed that individuals whose primary source of happiness is work/money/others are less likely to report being "very happy/happy" compared to those whose source is love, the reference category. Those whose primary source is work/money/others are more likely to be "moderately happy" than the reference group. Older individuals whose primary source of happiness is love have a higher probability of happiness compared to those whose source is work or money. Similar studies have identified emotional support, such as love, as having the most substantial impact on older individuals' quality of life and happiness levels [1, 112]. Positive affection is also a significant psychological factor in promoting and enhancing health behaviors [113].

Similarly, a study conducted in Portugal revealed a connection between happiness and mental health [114], suggesting that happiness and mental health are intertwined, protecting older populations from stress and helping them cope with problems [115]. This issue could play a crucial role in the decrease in happiness and psychological well-being among older individuals [116]. However, contrary to this study, other research indicates that socioeconomic status (including work, money, etc.) and other social characteristics could be directly associated with happiness [117].

The happiness of older individuals is also related to their level of hope. Hopeful older individuals are more likely to report being "very happy/happy" compared to their non-hopeful counterparts. They are also less likely to report being "moderately happy" compared to those without hope. Similar studies found that the hope levels significantly affect older individuals' happiness [17, 118, 119]. One study determined that the COVID-19 pandemic affected hope

levels, with older individuals who spent time with family members experiencing increased hope and happiness [17]. Another study determined that older individuals with high levels of hope also reported high levels of happiness [63]. In contrast, some studies also found high levels of hopelessness [120, 121].

Health satisfaction is another significant variable associated with the happiness levels of older individuals in this study. Those satisfied with their health are more likely to report being "very happy/happy" compared to those who are not satisfied. Conversely, those satisfied with their health are less likely to report being "moderately happy" compared to those who are not satisfied. Previous studies have shown that health perception affects happiness levels [47, 122]. An earlier study emphasized that health (physical, cognitive, etc.) is crucial to successful aging and happiness [58]. Access to healthcare services is another factor influencing older individuals' health. Primary caregivers play a significant role in physical health and are the primary source of mental health care for older individuals [123, 124]. Previous studies have shown that access to health services reduces mental disorders and increases happiness among older individuals [125–127].

This study also found that homeownership among older individuals is related to their happiness levels. Homeowners are more likely to report being "very happy/happy" compared to those who rent. Conversely, homeowners are less likely to report being "moderately happy" compared to renters. Similar studies have demonstrated that financial status affects life satisfaction [51, 88, 128]. A study among among older individuals in Türkiye determined that those with lower incomes are more likely to be unhappy, as having sufficient funds to cover material needs directly affects happiness [129]. In contrast, some studies have shown no significant relationship between income and happiness [3, 48].

The findings of this study provide insights into factors associated with happiness, which can inform policies in relevant areas. Policymakers can use these findings to develop programs that promote the happiness of older individuals. Understanding the results and which life factors have a greater or lesser impact on the perception of happiness can aid in the improvement efforts of service providers and specialists.

This study demonstrates that, in order to achieve happiness among older individuals in Türkiye, it is necessary to examine and further investigate several factors in samples with diverse sociodemographic characteristics. It is thought that this study will contribute to our understanding of the factors that affect the happiness levels of older individuals in Türkiye and may inspire additional research on the subject.

## Conclusions

According to the results, gender, age, educational status, source of happiness, health satisfaction, hope scale, and homeownership status were identified as significant determinants of happiness among older individuals during the COVID-19 pandemic.

The literature on happiness illustrates the relationship between various factors and happiness. Older individuals can participate in more entrepreneurial and educational activities through the development of programs. Due to the positive effect of an increase in income on happiness, the retirement period for older individuals can be enhanced, thereby contributing to an improved standard of living.

Society's emotional and psychological support for older individuals gives them hope for the future and may indicate rising happiness levels. In this case, providing older individuals with the necessary emotional and instrumental conditions may increase their happiness by easing their concerns about the pandemic period.

The community and professionals must implement corresponding interventions for older individuals to reduce the psychological damage and consequences caused by the pandemic crisis. During the COVID-19 pandemic, it may benefit older individuals to implement programs that increase hope, strengthen emotional support, and provide quality healthcare. Telephone and video conference calls can increase the life satisfaction of older individuals who cannot see their loved ones during this time. Additionally, older individuals should be provided with emotional support and opportunities for social engagement to promote their happiness. The negative effects of such situations as the COVID-19 pandemic can be mitigated through social organizations, healthcare providers, the media, and charities. Social environments in which they can interact, thereby expanding their social networks, should be created, and activities should be expanded.

Implementing initiatives that increase hope for older individuals, strengthen emotional support, and provide quality health care may benefit the lives of those who have retired and deserve a peaceful, comfortable old age. Telephone and video conference calls for older individuals who cannot meet with their loved ones during this period can help increase their happiness levels.

Determinants of happiness can vary under different living conditions. Therefore, this study can be replicated with older participants. Further studies are needed to determine whether changes and interventions in policies addressing older individuals affect their quality of life and happiness. The present study is desired to inspire more scientists to focus on the happiness of older individuals.

## Limitations of the study

This research has some limitations. Participants in the Life Satisfaction Survey conducted by the Turkish Statistical Institute during COVID-19 were not asked if they had coronavirus disease. There is no information about the participants' coronavirus disease. Therefore, it was impossible to differentiate between infected and uninfected participants. The data in this study are secondary data. The variables required for statistical analysis consist of the variables in the data set. However, the analysis did not include variables that were not included in the data set. The conclusions were limited to the factors studied in the study. While the study provides insights into the factors affecting the happiness of older individuals in Turkey during the COVID-19 pandemic, its findings may not directly apply to other cultural or demographic contexts. The data regarding the questions asked to individuals to determine their happiness levels are the individuals' responses. The study relies on self-reported data from the Life Satisfaction Survey, which may introduce bias due to participants' subjective interpretations of their happiness levels. Therefore, the data obtained through the data collection method may be biased. Since it is survey data, methodology or data analysis may be biased. In addition, since the data are cross-sectional, the definite causal relationship between happiness and related socio-economic factors cannot be inferred. We propose further longitudinal research.

## Acknowledgments

The authors would like to thank the Turkish Statistical Institute for the data. The views and opinions expressed in this manuscript are those of the authors only and do not necessarily represent the views, official policy, or position of the Turkish Statistical Institute.

## Author Contributions

**Conceptualization:** Ahmet Kamil Kabakuş, Ömer Alkan.

**Formal analysis:** Ömer Alkan.

**Investigation:** Nurşen Çomaklı Duvar, Neslihan İyit.

**Methodology:** Ömer Alkan.

**Resources:** Nurşen Çomaklı Duvar, Ahmet Kamil Kabakuş, Neslihan İyit.

**Software:** Ahmet Kamil Kabakuş.

**Validation:** Nurşen Çomaklı Duvar.

**Writing – original draft:** Nurşen Çomaklı Duvar, Ömer Alkan.

**Writing – review & editing:** Nurşen Çomaklı Duvar, Ahmet Kamil Kabakuş, Neslihan İyit, Ömer Alkan.

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
