## [Decision Letter · Decision Letter 0]

22 May 2024

PONE-D-23-40834A Study on the Determination of the Factors Affecting the Happiness Levels of Older Individuals During the COVID-19 Pandemic in Turkish SocietyPLOS ONE

Dear Dr. Alkan,

Thank you for submitting your manuscript to PLOS ONE. After careful consideration, we feel that it has merit but does not fully meet PLOS ONE’s publication criteria as it currently stands. Therefore, we invite you to submit a revised version of the manuscript that addresses the points raised during the review process.

We look forward to receiving your revised manuscript.

Kind regards,

Boshra Ismael Ahmed Arnout

Academic Editor

PLOS ONE

Journal Requirements:

2. In the online submission form, you indicated that your data is available only on request from a third party. Please note that your Data Availability Statement is currently missing [the contact details for the third party, such as an email address or a link to where data requests can be made]. Please update your statement with the missing information. 

Reviewers' comments:

Reviewer's Responses to Questions

**Comments to the Author**

1. Is the manuscript technically sound, and do the data support the conclusions?

Reviewer #1: Partly

Reviewer #2: No

Reviewer #3: Yes

2. Has the statistical analysis been performed appropriately and rigorously? 

Reviewer #1: Yes

Reviewer #2: No

Reviewer #3: N/A

3. Have the authors made all data underlying the findings in their manuscript fully available?

Reviewer #1: Yes

Reviewer #2: No

Reviewer #3: Yes

4. Is the manuscript presented in an intelligible fashion and written in standard English?

Reviewer #1: No

Reviewer #2: No

Reviewer #3: Yes

5. Review Comments to the Author

**Reviewer #1:** I am pleased that you have invited me to review a manuscript titled “A Study on the Determination of the Factors Affecting the Happiness Levels of Older Individuals During the COVID-19 Pandemic in Turkish Society”. The manuscript is easy to read and understand. Also, the manuscript is nicely presented, following a thorough reading and comprehension of the paper, I forwarded the following comments.

1. There are numerous typographical and grammatical issues that should be handled.

2. In the abstract section “1,863 individuals aged 60 and participated in this survey”. Write as 60 and above.

3. From lines 257-260 the manuscript mentions about chi-square test and it says, that the test not only determines the significance of observed differences but also provides information about the specific categories where differences occur. Is that true? The chi-square test didn’t provide any information about in which category the differences occur. It should be corrected.

4. Regarding Education Level, the categories are not exhaustive. What about those who have a Diploma, and TVET colleges’ certificate, MSc and above? What does that mean no degree?

5. Categories of some variables are ambiguous. If we see the variable “Source of Happiness as a Person”: (Himself/herself/spouse, Children/mother/father, Friends/ nephews/ grandchildren/ other, All family members), and

• “Source of Happiness”: (Success, Job/money/other, Health, Love), you have used the word other. This leads to confusion. Because if we use this term (other), all other categories are also included which leads miss conception. Should take measures.

6. Even if in line 305 you expressed the absence of a multicollinearity problem between the variables, variables like Hope for the future and Hope full scale are nearly similar and redundant variables. In addition frequency for neither hopeful nor not hopeful is missed. So better if you use either variable only.

7. In lines 336-337, it says that “Individuals with very satisfied/satisfied health satisfaction are 10.8% less likely to be “moderately happy” in the reference category”. What does that mean? Re-write it. Similarly for a statement on lines 339-340, “The probability of being “very happy/happy is 19.6% and 22.2% lower than the hopeful reference category”

8. The major issue I saw in this study is the lack of justification. For example, the results of the study state that “men are less likely to be very happy/happy than women. …. Contrary to this study, some studies concluded that men are happier than women [34, 75]”. This needs justification as to why men are less likely to be “very happy/happy” than women and why the results of this study contradict with other studies done in different places at different times. Similarly for all the rest variables.

**Reviewer #2: **Referee report for “A Study on the Determination of the Factors Affecting the Happiness Levels of Older Individuals During the COVID-19 Pandemic in Turkish Society”

Summary

The study aimed to identify factors influencing the happiness levels of older individuals in Turkey during the COVID-19 pandemic. It utilized data from the 2020 Life Satisfaction Survey conducted by TURKSTAT, involving 1,863 participants aged 60 and above. The relationship between happiness levels and various factors was analyzed using the chi-square independence test and generalized ordered logistic regression analysis. Results showed that individuals aged 60-64 were 10.1% less likely to be happy compared to those aged 65 and older, while men were 4.3% less likely to be happy than women. Moreover, individuals with lower education levels had higher probabilities of happiness, and those hopeful about the future were less likely to be happy. The study identified gender, age, education, source of happiness, health satisfaction, hope scale, and homeownership as factors impacting happiness levels in older individuals, suggesting societal support as an indicator of happiness.

Comments to author

Major issues:

1. The introduction provides a general overview of happiness and its importance but lacks a clear statement of the research objectives. The abstract and introduction should clearly outline the aim of the study, which seems to be identifying factors influencing the happiness levels of older individuals during the COVID-19 pandemic in Turkey. However, this should be explicitly stated.

2. While the study mentions various factors affecting happiness, it lacks a theoretical framework that would provide a structured understanding of how these factors interact and influence happiness levels among older individuals during the pandemic.

3. The methods section lacks detail on certain crucial aspects, such as the selection criteria for the sample and the specific measures used to assess happiness levels. Additionally, the description of the statistical analysis could be more detailed to provide clarity on the modeling approach employed.

4. The study relies on self-reported data from the Life Satisfaction Survey, which may introduce bias due to participants' subjective interpretations of their happiness levels.

5. The study's cross-sectional design limits its ability to establish causal relationships between happiness and socio-economic factors. It acknowledges this limitation but does not propose avenues for further longitudinal research.

6. The study lacks information on participants' COVID-19 status, which could be crucial in understanding the pandemic's direct impact on happiness levels.

7. While the study provides insights into the factors affecting the happiness of older individuals in Turkey during the COVID-19 pandemic, its findings may not be directly applicable to other cultural or demographic contexts.

8. While the study employs statistical methods like chi-square tests and logistic regression, it lacks detailed discussion on the assumptions underlying these techniques and potential limitations.

9. While the study presents regression coefficients and odds ratios, there's a lack of detailed interpretation of these results. Providing more context and discussing the practical implications of the findings would enrich the discussion section.

Minor issues:

10. There are minor typographical errors, such as missing punctuation marks (e.g., comma after "In addition" in line 202) and inconsistencies in capitalization (e.g., "Türkiye" vs. "TURKSTAT"). Proofreading for such errors would enhance the readability of the manuscript.

11. Certain sections could be clearer, especially in explaining the rationale behind the choice of variables and the interpretation of results.

12. The discussion section briefly touches on each variable but could benefit from a deeper exploration of the literature and theoretical frameworks supporting the study's findings.

13. There are occasional grammatical errors and awkward phrasings that could be smoothed out for clarity and readability.

14. The study does not explicitly mention any limitations or potential biases in the methodology or data analysis. Acknowledging these limitations would provide transparency and help readers interpret the findings more accurately.

15. The study briefly mentions the need for further research but could provide more specific suggestions for future studies, such as exploring longitudinal data or conducting qualitative research to complement quantitative findings.

**Reviewer #3:** Dear Authors,

Thank you for your efforts in revising the manuscript titled "A Study on the Determination of the Factors Affecting the Happiness Levels of Older Individuals During the COVID-19 Pandemic in Turkish Society."

Overall Analysis:

The manuscript analyzes data collected by TURKSTAT in 2020, focusing on factors influencing the happiness levels of elderly individuals in Türkiye during the COVID-19 pandemic. This analysis offers valuable insight into the socio-psychological trends among Turkish seniors during this challenging period. While the studied factors have been explored previously in both Turkish and non-Turkish populations, your focus on the pandemic sheds light on the unique struggles, concerns, and challenges faced by older adults during this stressful time.

Minor Suggestions:

1. Data Subgroup Analysis: While I am not a statistical expert, I recommend considering dividing the data of 1863 individuals into age groups (e.g., 60-69, 70-79, 80-89) for a more precise analysis of the studied factors. This would allow for a nuanced understanding of happiness across different age segments within the elderly population.

2. Happiness Factors by Region: In the discussion of happiness factors, consider categorizing references geographically (e.g., continents, developed vs. developing countries) to highlight potential similarities or differences based on socio-economic characteristics. This would provide a more structured approach to the existing analysis.

3. Reference Refinement: In lines 129-132, you mention research conducted in the Americas and South Africa. While both regions fall under the Americas classification, there might be significant cultural and socio-economic differences. For a more focused comparison, consider referencing research specific to South Africa or separately addressing research from the Americas and South Africa.

Additionally, I suggest Hou et al. (2023) as a potential reference – a valuable addition to support the discussion on religious belief and happiness.

Hou B, Wu Y and Huang Y. How Does Religious Belief Affect Happiness of Older Adults: The Participation Effect and Support Effect. Religions.2023; 14: 243. https://doi.org/10.3390/rel14020243

4. Explaining Results: In the discussion section, consider providing explanations for some of the findings. For example, line 370-371 suggests individuals with less education might be happier than those with a university degree. Including a possible explanation for this result would strengthen your analysis.

The same approach could be applied to other unexpected findings to enrich the discussion.

Thank you again for your work. These suggestions aim to further enhance the clarity and impact of your manuscript.

Sincerely,

6. PLOS authors have the option to publish the peer review history of their article (what does this mean?). If published, this will include your full peer review and any attached files.

Reviewer #1: **Yes: **Seid Ali Tareke

Reviewer #2: **Yes: **Bibhuti Sarker

Reviewer #3: **Yes: **Assist. Prof. Luma H. A. Al Obaidy

---

## [Author Response · Author response to Decision Letter 0]

15 Jul 2024

Dear Editor and Reviewers,

Thank you very much for your comments concerning our manuscript titled “A Study on the Determination of the Factors Affecting the Happiness Levels of Older Individuals During the COVID-19 Pandemic in Turkish Society”. These comments have been very helpful in reviewing and improving our manuscript. We have carefully revising these instructive comments and made corrections that we hope will be approved. The revised parts are highlighted in red in the main paper.

The corrections to the manuscript and responses to the reviewer’s comments are as following:

Regards,

Authors

Review Comments to the Author

Please use the space provided to explain your answers to the questions above. You may also include additional comments for the author, including concerns about dual publication, research ethics, or publication ethics. (Please upload your review as an attachment if it exceeds 20,000 characters).

Reviewer #1: 

I am pleased that you have invited me to review a manuscript titled “A Study on the Determination of the Factors Affecting the Happiness Levels of Older Individuals During the COVID-19 Pandemic in Turkish Society”. The manuscript is easy to read and understand. Also, the manuscript is nicely presented, following a thorough reading and comprehension of the paper, I forwarded the following comments.

Reviewer point #1: There are numerous typographical and grammatical issues that should be handled.

Author response #1: Thank you for the comment. Language and grammatical errors were corrected during the “proofreading” made by the language editor.

Reviewer point #2: In the abstract section “1,863 individuals aged 60 and participated in this survey”. Write as 60 and above.

Author response #2: Thank you for the comment. Taking this criticism into account, we have revised the statement.

Reviewer point #3: From lines 257-260 the manuscript mentions about chi-square test and it says, that the test not only determines the significance of observed differences but also provides information about the specific categories where differences occur. Is that true? The chi-square test didn’t provide any information about in which category the differences occur. It should be corrected.

Author response #3: Thank you for the comment. Taking this criticism into account, we have revised the sentence.

Reviewer point #4: Regarding Education Level, the categories are not exhaustive. What about those who have a Diploma, and TVET colleges’ certificate, MSc and above? What does that mean no degree?

Author response #4: Thank you for the comment. Taking this criticism into account, we separated the categories on the education level variable and re-estimated the model. We reinterpreted the results according to the new model. Changed to “did not complete a school”.

Reviewer point #5: Categories of some variables are ambiguous. If we see the variable “Source of Happiness as a Person”: (Himself/herself/spouse, Children/mother/father, Friends/ nephews/ grandchildren/ other, All family members), and “Source of Happiness”: (Success, Job/money/other, Health, Love), you have used the word other. This leads to confusion. Because if we use this term (other), all other categories are also included which leads miss conception. Should take measures.

Author response #5: Thank you for the comment. Taking this criticism into account, we have revised the statement.

Reviewer point #6: Even if in line 305 you expressed the absence of a multicollinearity problem between the variables, variables like Hope for the future and Hope full scale are nearly similar and redundant variables. In addition frequency for neither hopeful nor not hopeful is missed. So better if you use either variable only.

Author response #6: Thank you for the comment. Taking this criticism into account, we removed the Hope for the future variable and re-estimated the model. We reinterpreted the results according to the new model.

Reviewer point #7: In lines 336-337, it says that “Individuals with very satisfied/satisfied health satisfaction are 10.8% less likely to be “moderately happy” in the reference category”. What does that mean? Re-write it. Similarly for a statement on lines 339-340, “The probability of being “very happy/happy is 19.6% and 22.2% lower than the hopeful reference category”.

Author response #7: Thank you for the comment. Taking this criticism into account, we have revised the statement.

Reviewer point #8: The major issue I saw in this study is the lack of justification. For example, the results of the study state that “men are less likely to be very happy/happy than women. …. Contrary to this study, some studies concluded that men are happier than women [34, 75]”. This needs justification as to why men are less likely to be “very happy/happy” than women and why the results of this study contradict with other studies done in different places at different times. Similarly for all the rest variables.

Author response #8: Thank you for the comment. Taking this criticism into account, we have revised the discussion. The discussion has been rewritten. 

Reviewer #2: 

Referee report for “A Study on the Determination of the Factors Affecting the Happiness Levels of Older Individuals During the COVID-19 Pandemic in Turkish Society”

Summary

The study aimed to identify factors influencing the happiness levels of older individuals in Turkey during the COVID-19 pandemic. It utilized data from the 2020 Life Satisfaction Survey conducted by TURKSTAT, involving 1,863 participants aged 60 and above. The relationship between happiness levels and various factors was analyzed using the chi-square independence test and generalized ordered logistic regression analysis. Results showed that individuals aged 60-64 were 10.1% less likely to be happy compared to those aged 65 and older, while men were 4.3% less likely to be happy than women. Moreover, individuals with lower education levels had higher probabilities of happiness, and those hopeful about the future were less likely to be happy. The study identified gender, age, education, source of happiness, health satisfaction, hope scale, and homeownership as factors impacting happiness levels in older individuals, suggesting societal support as an indicator of happiness.

Comments to author

Major issues:

Reviewer point #1: The introduction provides a general overview of happiness and its importance but lacks a clear statement of the research objectives. The abstract and introduction should clearly outline the aim of the study, which seems to be identifying factors influencing the happiness levels of older individuals during the COVID-19 pandemic in Turkey. However, this should be explicitly stated.

Author response #1: Thank you for the comment. Taking this criticism into account, we have added the necessary explanations in introduction.

Reviewer point #2: While the study mentions various factors affecting happiness, it lacks a theoretical framework that would provide a structured understanding of how these factors interact and influence happiness levels among older individuals during the pandemic.

Author response #2: Thank you for the comment. Taking this criticism into account, we have revised the introduction. We have added the necessary explanations in introduction.

Reviewer point #3: The methods section lacks detail on certain crucial aspects, such as the selection criteria for the sample and the specific measures used to assess happiness levels. Additionally, the description of the statistical analysis could be more detailed to provide clarity on the modeling approach employed.

Author response #3: Thank you for the comment. Taking this criticism into account, we have revised the Data and Statistical Analysis sections. We have added the Figure 1 for for the sample.

Reviewer point #4: The study relies on self-reported data from the Life Satisfaction Survey, which may introduce bias due to participants' subjective interpretations of their happiness levels.

Author response #4: Thank you for the comment. Taking this criticism into account, we have revised the Limitations of the study.

Reviewer point #5: The study's cross-sectional design limits its ability to establish causal relationships between happiness and socio-economic factors. It acknowledges this limitation but does not propose avenues for further longitudinal research.

Author response #5: Thank you for the comment. Taking this criticism into account, we have revised the Limitations of the study.

Reviewer point #6: The study lacks information on participants' COVID-19 status, which could be crucial in understanding the pandemic's direct impact on happiness levels.

Author response #6: Thank you for the comment. The survey did not ask participants about their COVID-19 status. This limitation was stated in the study. Taking this criticism into account, we have revised the Limitations of the study.

Reviewer point #7: While the study provides insights into the factors affecting the happiness of older individuals in Turkey during the COVID-19 pandemic, its findings may not be directly applicable to other cultural or demographic contexts.

Author response #7: Thank you for the comment. Taking this criticism into account, we have revised the Limitations of the study.

Reviewer point #8: While the study employs statistical methods like chi-square tests and logistic regression, it lacks detailed discussion on the assumptions underlying these techniques and potential limitations.

Author response #8: Thank you for the comment. Taking this criticism into account, we have revised the Data and Statistical Analysis sections.

Reviewer point #9: While the study presents regression coefficients and odds ratios, there's a lack of detailed interpretation of these results. Providing more context and discussing the practical implications of the findings would enrich the discussion section.

Author response #9: Thank you for the comment. We interpreted the marginal effects. Taking this criticism into account, we have revised the discussion. The discussion has been rewritten. 

Minor issues:

Reviewer point #10: There are minor typographical errors, such as missing punctuation marks (e.g., comma after "In addition" in line 202) and inconsistencies in capitalization (e.g., "Türkiye" vs. "TURKSTAT"). Proofreading for such errors would enhance the readability of the manuscript.

Author response #10: Thank you for the comment. Language and grammatical errors were corrected during the “proofreading” made by the language editor. 

The United Nations has decided to use “Türkiye” instead of “Turkey” internationally as of June 2022. Due to this decision, the word “Türkiye” was used in the manuscript.

(https://turkiye.un.org/en/184798-turkeys-name-changed-t%C3%BCrkiye)

Reviewer point #11: Certain sections could be clearer, especially in explaining the rationale behind the choice of variables and the interpretation of results.

Author response #11: Thank you for the comment. Taking this criticism into account, we have revised the results.

Reviewer point #12: The discussion section briefly touches on each variable but could benefit from a deeper exploration of the literature and theoretical frameworks supporting the study's findings.

Author response #12: Thank you for the comment. Taking this criticism into account, we have revised the discussion. The discussion has been rewritten.

Reviewer point #13: There are occasional grammatical errors and awkward phrasings that could be smoothed out for clarity and readability.

Author response #13: Thank you for the comment. Language and grammatical errors were corrected during the “proofreading” made by the language editor. 

Reviewer point #14: The study does not explicitly mention any limitations or potential biases in the methodology or data analysis. Acknowledging these limitations would provide transparency and help readers interpret the findings more accurately.

Author response #14: Thank you for the comment. Taking this criticism into account, we have revised the Data and Statistical Analysis sections. 

Reviewer point #15: The study briefly mentions the need for further research but could provide more specific suggestions for future studies, such as exploring longitudinal data or conducting qualitative research to complement quantitative findings.

Author response #15: Thank you for the comment. Taking this criticism into account, we have added the necessary explanations in conslucios.

Reviewer #3: 

Dear Authors,

Thank you for your efforts in revising the manuscript titled "A Study on the Determination of the Factors Affecting the Happiness Levels of Older Individuals During the COVID-19 Pandemic in Turkish Society."

Overall Analysis:

The manuscript analyzes data collected by TURKSTAT in 2020, focusing on factors influencing the happiness levels of elderly individuals in Türkiye during the COVID-19 pandemic. This analysis offers valuable insight into the socio-psychological trends among Turkish seniors during this challenging period. While the studied factors have been explored previously in both Turkish and non-Turkish populations, your focus on the pandemic sheds light on the unique struggles, concerns, and challenges faced by older adults during this stressful time.

Minor Suggestions:

Reviewer point #1: Data Subgroup Analysis: While I am not a statistical expert, I recommend considering dividing the data of 1863 individuals into age groups (e.g., 60-69, 70-79, 80-89) for a more precise analysis of the studied factors. This would allow for a nuanced understanding of happiness across different age segments within the elderly population.

Author response #1: Thank you for the comment. Taking this criticism into account, we separated the categories on the age variable and re-estimated the model. We reinterpreted the results according to the new model.

Reviewer point #2: Happiness Factors by Region: In the discussion of happiness factors, consider categorizing references geographically (e.g., continents, developed vs. developing countries) to highlight potential similarities or differences based on socio-economic characteristics. This would provide a more structured approach to the existing analysis.

Author response #2: Thank you for the comment. Taking this criticism into account, we have revised the Literature.

Reviewer point #3: Reference Refinement: In lines 129-132, you mention research conducted in the Americas and South Africa. While both regions fall under the Americas classification, there might be significant cultural and socio-economic differences. For a more focused comparison, consider referencing research specific to South Africa or separately addressing research from the Americas and South Africa.

Additionally, I suggest Hou et al. (2023) as a potential reference – a valuable addition to support the discussion on religious belief and happiness.

Hou B, Wu Y and Huang Y. How Does Religious Belief Affect Happiness of Older Adults: The Participation Effect and Support Effect. Religions.2023; 14: 243. https://doi.org/10.3390/rel14020243

Author response #3: Thank you for the comment. Taking this criticism into account, we have revised the Literature and Discussion. We examined the relevant references in detail.

Reviewer point #4: Explaining Results: In the discussion section, consider providing explanations for some of the findings. For example, line 370-371 suggests individuals with less education might be happier than those with a university degree. Including a possible explanation for this result would strengthen your analysis.

The same approach could be applied to other unexpected findings to enrich the discussion.

Thank you again for your work. These suggestions aim to further enhance the clarity and impact of your manuscript.

Sincerely,

Author response #4: Thank you for the comment. Taking this criticism into account, we have revised the Results and Discussion.

---

## [Decision Letter · Decision Letter 1]

14 Oct 2024

PONE-D-23-40834R1A Study on the Determination of the Factors Affecting the Happiness Levels of Older Individuals During the COVID-19 Pandemic in Turkish SocietyPLOS ONE

Dear Dr. Alkan,

Thank you for submitting your manuscript to PLOS ONE. After careful consideration, we feel that it has merit but does not fully meet PLOS ONE’s publication criteria as it currently stands. Therefore, we invite you to submit a revised version of the manuscript that addresses the points raised during the review process.

We look forward to receiving your revised manuscript.

Kind regards,

Boshra A Arnout

Academic Editor

PLOS ONE

Reviewers' comments:

Reviewer's Responses to Questions

**Comments to the Author**

1. If the authors have adequately addressed your comments raised in a previous round of review and you feel that this manuscript is now acceptable for publication, you may indicate that here to bypass the “Comments to the Author” section, enter your conflict of interest statement in the “Confidential to Editor” section, and submit your "Accept" recommendation.

Reviewer #3: All comments have been addressed

Reviewer #4: (No Response)

2. Is the manuscript technically sound, and do the data support the conclusions?

Reviewer #3: Yes

Reviewer #4: Yes

3. Has the statistical analysis been performed appropriately and rigorously? 

Reviewer #3: Yes

Reviewer #4: Yes

4. Have the authors made all data underlying the findings in their manuscript fully available?

Reviewer #3: Yes

Reviewer #4: No

5. Is the manuscript presented in an intelligible fashion and written in standard English?

Reviewer #3: Yes

Reviewer #4: Yes

6. Review Comments to the Author

Reviewer #3: To the authors,

Thank you for completing the required amendments in the review and I would like to point out that the conclusions should be limited to the factors studied in the study.

Best wishes

Reviewer #4: Thank you for the opportunity to review your manuscript. The study examined the factors influencing happiness levels among different demographic groups in Turkey during the COVID-19 pandemic. While I appreciate the efforts put into conducting this research, I have observed significant overlaps in the overarching themes, methodologies, and some aspects of the analysis with another recently published work, by some of the same authors (https://doi.org/10.5455/PBS.20230512053548).

Such similarities raise concerns about the originality and incremental contribution of this manuscript. It is crucial for the advancement of scholarly work that each publication offers a distinct and significant contribution to the existing body of knowledge. In light of this, I would recommend clarifying how this manuscript distinctly advances our understanding compared to previously published works, especially those conducted using the same dataset and similar methodology.

In addition, I have a major concern about the reproducibility of the current findings. Although the raw data may be subject to third-party restrictions in reproduction, providing the statistical scripts and models used in the research would enhance transparency and verifiability. This is crucial for allowing future researchers to validate and reproduce your findings and facilitates comparative studies in cross-cultural or similar contexts. I recommend including the scripts for data preprocessing and analysis in the supplementary materials.

Thank you for addressing these concerns.

7. PLOS authors have the option to publish the peer review history of their article (what does this mean?). If published, this will include your full peer review and any attached files.

Reviewer #3: **Yes: **Luma Hassan Alwan Al Obaidy

Reviewer #4: No

---

## [Author Response · Author response to Decision Letter 1]

15 Oct 2024

Dear Editor and Reviewers,

Thank you very much for your comments concerning our manuscript titled “A Study on the Determination of the Factors Affecting the Happiness Levels of Older Individuals During the COVID-19 Pandemic in Turkish Society”. These comments have been very helpful in reviewing and improving our manuscript. We have carefully revising these instructive comments and made corrections that we hope will be approved. The revised parts are highlighted in red in the main paper.

The corrections to the manuscript and responses to the reviewer’s comments are as following:

Regards,

Authors

Reviewer #3: 

To the authors,

Reviewer point #1: Thank you for completing the required amendments in the review and I would like to point out that the conclusions should be limited to the factors studied in the study.

Best wishes

Author response #1: Thank you for the comment. Taking this criticism into account, we have added “The conclusions were limited to the factors studied in the study” to the Limitations of the study section.

Reviewer #4: 

Reviewer point #1: Thank you for the opportunity to review your manuscript. The study examined the factors influencing happiness levels among different demographic groups in Turkey during the COVID-19 pandemic. While I appreciate the efforts put into conducting this research, I have observed significant overlaps in the overarching themes, methodologies, and some aspects of the analysis with another recently published work, by some of the same authors (https://doi.org/10.5455/PBS.20230512053548).

Such similarities raise concerns about the originality and incremental contribution of this manuscript. It is crucial for the advancement of scholarly work that each publication offers a distinct and significant contribution to the existing body of knowledge. In light of this, I would recommend clarifying how this manuscript distinctly advances our understanding compared to previously published works, especially those conducted using the same dataset and similar methodology.

Author response #1: Thank you for the comment. In this study, data from elderly individuals over 60 years of age, one of the disadvantaged groups during the COVID-19 pandemic period, were used. Taking this criticism into account, we have revised the Introduction. We have added the following statements to the introduction.

“The older adults are at-risk groups during the COVID-19 pandemic. Identifying the factors affecting the happiness levels of older individuals during the COVID-19 period is of great importance for developing more targeted strategies to minimize the adverse effects of the pandemic on the elderly and protect their quality of life. It is important to improve the quality of life of older individuals and provide a better ageing process. COVID-19 has been a disease that carries serious health risks for older individuals. The uncertainties of COVID-19 and health concerns have led to an increase in mental health problems such as anxiety, depression and stress in older adults. To cope with these problems, it is critical to identify the factors that affect happiness and adapt mental health services accordingly. Therefore, identifying the factors that affect the happiness levels of older individuals to protect their physical and psychological health may help reduce the disease's adverse effects.

The findings indicate that several factors affect the happiness levels of those aged 60 and older. According to the research, factors such as gender, age, education level, income level, and being hopeful affect the happiness of older individuals. It is essential to know these factors to design more effective social policy and health services according to the needs of the elderly population. Thus, better health services, social assistance and education programs can be developed for older adults. To better understand the relationship between happiness-related factors, a larger study sample is needed among older individuals whose lives were changed by strict isolation methods during the COVID-19 period in Türkiye. For this reason, the purpose of this study is to review the current policies throughout the country, to re-evaluate the needs that arise in the face of extraordinary altering circumstances such as the COVID-19 pandemic, and to fill this gap in the literature. 

Understanding the factors affecting the happiness levels of older individuals during the COVID-19 period can increase the effectiveness of the measures and support mechanisms to be taken to help them overcome this difficult period with less damage and live a healthier and happier old age.”

Reviewer point #2: In addition, I have a major concern about the reproducibility of the current findings. Although the raw data may be subject to third-party restrictions in reproduction, providing the statistical scripts and models used in the research would enhance transparency and verifiability. This is crucial for allowing future researchers to validate and reproduce your findings and facilitates comparative studies in cross-cultural or similar contexts. I recommend including the scripts for data preprocessing and analysis in the supplementary materials.

Thank you for addressing these concerns.

Author response #2: Thank you for the comment. Taking this criticism into account, we have revised the paper. We have added the following statements.

“The data were obtained through the joint teamwork of both the Turkish Statistical Institute (TSI) and the European Union Statistical Office (SOEU). We obtained this data from TSI in return for a contract without needing an ethics committee document and used it in our study. 

TSI is an institution that compiles, evaluates, and presents statistical information to decision-makers to prepare development plans and programs, make economic decisions, and address all other issues needed. TSI carries out internationally comparable statistical production activities according to the standards of organizations such as the European Union Statistical Office, the United Nations, OECD, ILO, etc. TSI collects data within the scope of the Official Statistics Program. The Official Statistics Program is prepared for five-year periods based on the Turkish Statistics Law No. 5429 to determine the basic principles and standards regarding the production and publication of official statistics and to ensure the production of up-to-date, reliable, timely, transparent and impartial data in areas of need at national and international levels [128]. TSI also conducts the Türkiye Health Survey within the scope of the Official Statistics Program put into effect by law. Since the Türkiye Health Survey is conducted within the scope of legal responsibility by the state, ethical approval is not required [129].

For this study, secondary data were employed. Official approval was received from the Turkish Statistical Institute to use the microdata set from the Life Satisfaction Survey. The Turkish Statistical Institute also received a “Letter of Undertaking” authorizing it to use the study’s data.

The letter of undertaking for the use of micro data without restrictions in dissemination:

Article 1- This letter of undertaking determines the rules, principles and obligations of the use of micro data, which are safe to disclose apart from the Presidency.

Article 2-This letter of undertaking regulates the use of micro data sets of Life Satisfaction Survey, 2020, within the framework of the Directive on Access and Use of Micro Data in line with the purpose specified in Article 1.

Article 3- The following provisions apply for the use of micro data:

a) Findings obtained by the researcher as a result of incorrect calculation only bind the researcher.

b) The researcher refers to the micro data of the Institution that he uses while disclosing the results obtained from the study.

c) The researcher is obliged to send a copy of the published report, article, publication etc. to the Institution Library within three months at the latest. Subsequent micro data usage requests of the researcher who is found not to fulfill this obligation are not covered.

d) The researcher cannot reproduce, give to third parties, sell or transfer the micro data set he obtained.

Article 4-The researcher, by taking into account the principles of confidentiality defined in 13. and 14. articles of Turkish Statistical Institution numbered 5429 and Regulation on Procedures and Principles Regarding Data Confidentiality and Confidential Data Security in Official Statistics, is deemed to guarantee hereby that he shall not disclose the information, table, etc. violating this principle and shall only use micro data for statistical purposes.”

---

## [Decision Letter · Decision Letter 2]

5 Dec 2024

A Study on the Determination of the Factors Affecting the Happiness Levels of Older Individuals During the COVID-19 Pandemic in Turkish Society

PONE-D-23-40834R2

Dear Dr. Alkan,

We’re pleased to inform you that your manuscript has been judged scientifically suitable for publication and will be formally accepted for publication once it meets all outstanding technical requirements.

Kind regards,

Boshra A. Arnout

Academic Editor

PLOS ONE

Additional Editor Comments (optional):

Reviewers' comments:

Reviewer's Responses to Questions

**Comments to the Author**

1. If the authors have adequately addressed your comments raised in a previous round of review and you feel that this manuscript is now acceptable for publication, you may indicate that here to bypass the “Comments to the Author” section, enter your conflict of interest statement in the “Confidential to Editor” section, and submit your "Accept" recommendation.

Reviewer #3: All comments have been addressed

Reviewer #5: All comments have been addressed

2. Is the manuscript technically sound, and do the data support the conclusions?

Reviewer #3: (No Response)

Reviewer #5: Yes

3. Has the statistical analysis been performed appropriately and rigorously? 

Reviewer #3: Yes

Reviewer #5: Yes

4. Have the authors made all data underlying the findings in their manuscript fully available?

Reviewer #3: Yes

Reviewer #5: Yes

5. Is the manuscript presented in an intelligible fashion and written in standard English?

Reviewer #3: Yes

Reviewer #5: Yes

6. Review Comments to the Author

Reviewer #3: (No Response)

Reviewer #5: While the study acknowledges its limitations, such as reliance on self-reported data and exclusion of certain demographics, a more thorough discussion of these limitations and their implications would enhance transparency. Additionally, potential biases related to the pandemic context (e.g., heightened anxiety or stress affecting responses) should be considered.

The study concludes that various socio-demographic and economic factors significantly affect the happiness levels of older individuals during the pandemic. This has practical implications for policymakers and mental health practitioners aiming to support this demographic. Recommendations for future research could include longitudinal studies to assess changes in happiness over time or qualitative studies to explore the lived experiences of older adults during the pandemic.

7. PLOS authors have the option to publish the peer review history of their article (what does this mean?). If published, this will include your full peer review and any attached files.

Reviewer #3: **Yes: **Dr. Luma Hassan Alwan Al Obaidy

Reviewer #5: No

---

## [Editor Report · Acceptance letter]

11 Jan 2025

PONE-D-23-40834R2 

PLOS ONE

Dear Dr. Alkan, 

I'm pleased to inform you that your manuscript has been deemed suitable for publication in PLOS ONE. Congratulations! Your manuscript is now being handed over to our production team.

Kind regards, 

on behalf of

Professor Boshra A. Arnout 

Academic Editor

PLOS ONE